# Stacked unsupervised learning with a network architecture found by supervised meta-learning

## Abstract

Stacked unsupervised learning (SUL) seems more biologically plausible than backpropagation, because learning is local to each layer. But SUL has fallen far short of backpropagation in practical applications, undermining the idea that SUL can explain how brains learn. Here we show an SUL algorithm that can perform completely unsupervised clustering of MNIST digits with comparable accuracy relative to unsupervised algorithms based on backpropagation. Our algorithm is exceeded only by self-supervised methods requiring training data augmentation by geometric distortions. The only prior knowledge in our unsupervised algorithm is implicit in the network architecture. Multiple convolutional "energy layers" contain a sum-of-squares nonlinearity, inspired by "energy models" of primary visual cortex. Convolutional kernels are learned with a fast minibatch implementation of the K-Subspaces algorithm. High accuracy requires preprocessing with an initial whitening layer, representations that are less sparse during inference than learning, and rescaling for gain control. The hyperparameters of the network architecture are found by supervised meta-learning, which optimizes unsupervised clustering accuracy. We regard such dependence of unsupervised learning on prior knowledge implicit in network architecture as biologically plausible, and analogous to the dependence of brain architecture on evolutionary history.

## 1 Introduction

Recently there has been renewed interest in the hypothesis that the brain learns through some version of the backpropagation algorithm [31]. This hypothesis runs counter to the neuroscience textbook account that local learning mechanisms, such as Hebbian synaptic plasticity, are the basis for learning by real brains. The concept of local learning has fallen out of favor because it has been far eclipsed by backpropagation in practical applications. This was not always the case. Historically, a popular approach to visual object recognition was to repeatedly stack a single-layer unsupervised learning module to generate a multilayer network, as exemplified by Fukushima's pioneering Neocognitron [11]. Stacked unsupervised learning (SUL) avoids the need for the backward pass of backpropagation, because learning is local to each layer.

In the 2000s, SUL was quite popular. There were attempts to stack diverse kinds of unsupervised learning modules, such as sparse coding [21], restricted Boltzmann machines [14, 29], denoising autoencoders [46], K-Means [7], and independent subspace analysis [25].

SUL managed to generate impressive-looking feature hierarchies that are reminiscent of the hierarchy of visual cortical areas. Stacking restricted Boltzmann machines yielded features that were sensitive to oriented edges in the first layer, eyes and noses in the second, and entire faces in the third layer [29]. Stacking three sparse coding layers yielded an intuitive feature hierarchy where higher layers were more selective to whole MNIST digits and lower layers were selective to small strokes [43]. Although these feature hierarchies are pleasing to the eye, they have not been shown to be effective

Submitted to 36th Conference on Neural Information Processing Systems (NeurIPS 2022). Do not distribute.

for visual object recognition, in spite of recent efforts to revive SUL using sparse coding [43, 6] and similarity matching [39].

Here we show an SUL algorithm that can perform unsupervised clustering of MNIST digits with high accuracy (2% error). The clustering accuracy is as good as the best unsupervised learning algorithms based on backpropagation. As far as we know, our accuracy is only exceeded by self-supervised methods that require training data augmentation or architectures with hand-designed geometric transformations. Such methods use explicit prior knowledge in the form of geometric distortions to aid learning.

Our network contains three convolutional energy layers inspired by energy models of primary visual cortex [1], which contain a sum-of-squares nonlinearity. Our energy layer was previously used by [16] in their independent subspace analysis (ISA) algorithm for learning complex cells. The kernels of our convolutional energy layers are trained by K-Subspaces clustering [45] rather than ISA. We also provide a novel minibatch algorithm for K-Subspaces learning.

After training, the first energy layer contains neurons that are selective for simple features but invariant to local distortions. These are analogous to the complex cells in energy models of the primary visual cortex [1]. The invariances are learned here rather than hand-designed, similar to previous work [16, 15]. We go further by stacking multiple energy layers. The second and third energy layers learn more sophisticated kinds of invariant feature selectivity. As mentioned above, feature hierarchies have previously been demonstrated for SUL. The novelty here is the learning of a feature hierarchy that is shown to be useful for pattern recognition.

In the special case that the sum-of-squares contains a single term, or equivalently the subspaces are restricted to be rank one, our convolutional energy layer reduces to a conventional convolutional layer. Accuracy worsens considerably, consistent with the idea that the energy layers are important for learning invariances. The energy layers contain an adaptive thresholding that allows representations to be less sparse for inference than for learning. This is also shown to be important for attaining high accuracy, as has been reported for other SUL algorithms [8, 24]. Representations are rescaled for gain control, and the energy layers are preceded by a convolutional whitening layer. These aspects of the network are also important for high accuracy.

The detailed architecture of our unsupervised network depends on subspace number and rank, kernel size, and sparsity. We use automatic tuning software [2] to systematically search for a hyperparameter configuration that optimizes the clustering accuracy of the unsupervised network. Evaluating the clustering accuracy requires labeled examples, so the meta-learning is supervised, while the learning is unsupervised. A conceptually similar meta-learning approach has previously been applied to search for biologically plausible unsupervised learning algorithms [37].

For each iteration of the meta-learning, the weights of our network are initialized randomly before running the SUL algorithm. Therefore the only prior knowledge available to SUL resides in the architectural hyperparameters; no weights are retained from previous networks. We regard this implicit encoding of prior knowledge in network architecture as biologically plausible, because brain architecture also contains prior knowledge gained during evolutionary history, and meta-learning is analogous to biological evolution. In its own "lifetime" our network is able to learn with no labels at all. This is possible because the network is "born" with an architecture inherited from networks that "lived" previously.

## 2    Related work

**Independent subspace analysis** Our method is related to past works on independent subspace analysis [16, 17, 26], mixtures of principal components analyzers [13], and subspace clustering [45, 47]. A core idea behind these works is that invariances can be represented via subspaces. The most similar of these works to ours is [26] who stacked 2 layers of subspace features learned with independent subspace analysis to action recognition datasets.

**K-Means based features** Mathematically the work of [7, 9] is similar to ours. They use a variant of K-Means to learn patch features. Their learning algorithm (Algorithm 1) is a special case of our alg. 1 where they use 1D subspaces and full batch updates. Their inference procedure is also very similar, in that they use a dynamic threshold to sparsify patch vector representations. They use spatial pooling layers for invariance, whereas our pooling is learned by the energy layers. Their primary mode of

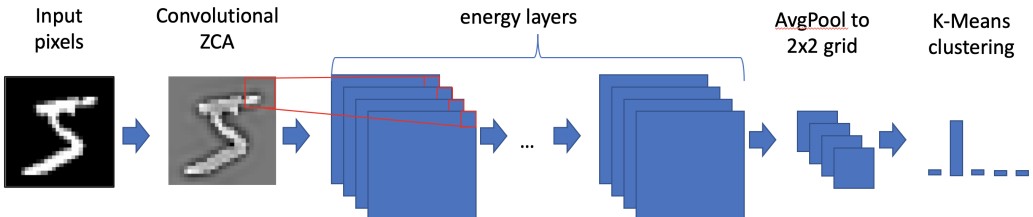

Figure 1: Our multilayer convolutional energy network. The last step of K-Means clustering can be regarded as part of the evaluation rather than the network itself.

evaluation was linear evaluation with the full labeled dataset, whereas we will seek to produce an unsupervised learning algorithm which clusters inputs without labels. They additionally employ a receptive field selection method.

**Capsule networks** Capsules are vector representations where the vector's length represents the probability of entity existence and the direction represents properties of that entity [41]. In our networks, the $r$-dimensional subspace vectors $\mathbf{Vx}$ can be interpreted as learned "pose" vectors. However our goal is to learn invariance, so we only propagate the norm $\|\mathbf{Vx}\|$, thus suppressing the pose details at each layer. Compared to the unsupervised capsule networks [23], our networks do not use any backpropagation of gradients, and do not rely on hand-designed affine transformations to generate representations.

**Meta-learning of unsupervised learning rules** Our work will rely on using a label-based clustering objective to evaluate and tune an unsupervised learning rule. In other words there are two levels of learning, an unsupervised inner loop and a supervised outer loop. This is the domain of meta-learning. The more common scenario for meta-learning is to focus on optimizing over a distribution of tasks, but for this work we will focus on one task. The work of [36] is perhaps the most closely related to ours. They use a few-shot supervised learning rule to tune an unsupervised learning algorithm that is a form of randomized backward propagation [30]. We wish to go further and remove any form of gradient feedback from a higher layer $L$ to a lower layer $L - 1$.

## 3   Network architecture

The overall network architecture is shown in Figure 1, and consists of a whitening layer followed by multiple convolutional energy layers and a final average pooling layer. The energy layers include adaptive thresholding to control sparsity of activity, as well as normalization of activity by rescaling.

**Convolutional ZCA** We define ZCA whitening for image patches in terms of the eigenvalues of the pixel-pixel correlation matrix. Our ZCA filter attenuates the top $k - 1$ eigenvalues, setting them equal to the $k$th largest eigenvalue. The smaller eigenvalues pass through unchanged. Our definition is slightly different from [10], which zeros out the smaller eigenvalues completely.

The first layer of our network is a convolutional variant of ZCA, in which each pixel of the output image is computed by applying ZCA whitening to the input patch centered at that location, and discarding all but the center pixel of the whitened output patch (p. 118 of [10]). The kernel has a single-pixel center with a diffuse surround (see Appendix). Patch size and number of whitened eigenvalues are specified in Table 1. Reflection padding is used to preserve the output size. Unsupervised learning algorithms such as sparse coding [40] and ICA [5] are often preceded by whitening when applied to images.

**Convolutional energy layer** We define the following modification of a convolutional layer, which computes $k$ output feature maps given $m$ input feature maps. We define the "feature vector" at a location to consist of the values of a set of feature maps at a given location.

1. Convolve the $m$ input feature maps with kernels to produce $kr$ feature maps ("S-maps") in the standard way, except with no bias or threshold.

2. Divide the $kr$ feature maps into $k$ groups of $r$. For each group, compute the Euclidean norm of the $r$-dimensional feature vector at each location. The result is $k$ feature maps ("C-maps").

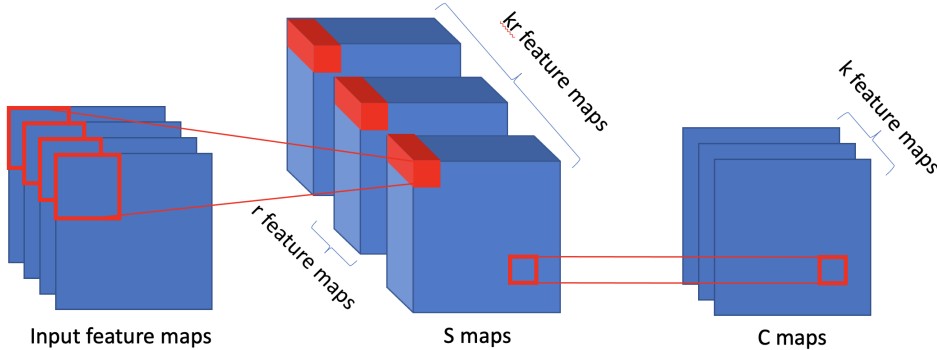

Figure 2: Diagram of a single energy layer. The outputs of this layer are the C feature maps, while S are just intermediate feature maps. We produce $kr$ feature maps (S) with a standard convolution. We produce (C) feature maps by square root the sum of squares inside each group of S feature maps, followed by an adaptive thresholding and rescaling.

Individual values in the S- and C-maps of Steps 1 and 2 will be called S-cells and C-cells, respectively. Each C-cell computes the square root of the sum-of-squares (Euclidean norm) of $r$ S-cells. The terms S-cells and C-cells are used in homage to [11]. They are reminiscent of energy models of primary visual cortex, in which complex cells compute the sum-of-squares of simple cell outputs [1].

The sum-of-squares can be regarded as a kind of pooling operation, but applied to S-cells at the same location in different feature maps. Pooling is typically performed on neighboring locations in the same feature map, yielding invariance to small translations by design. We will see later on that the sum-of-squares in the energy layer acquires invariances due to the learned kernels.

**Adaptive thresholding and normalization** It turns out to be important to postprocess the output of a convolutional energy layer as follows.

3. For each $k$-dimensional feature vector, adaptively threshold so that there are $W$ winners active.

4. Normalize the feature vector to unit Euclidean length, and rescale by multiplying with the Euclidean norm of the input patch at the same location.

Define $\mathbf{f}$ as the $k$-dimensional vector which is the $k$ feature map values at a location $u$ in the $C$ feature maps. In Step 3, the adaptive thresholding of $\mathbf{f}$ takes the form $\max\{0, f_i - \tau\}$ for $i = 1$ to $k$ where $\tau$ is the $W + 1$st largest element of $f$. Note that $\tau$ is set adaptively for each location. Such adaptive thresholding was used by [44] and is a version of the well-known $W$-winners-take-all concept [33]. After thresholding, C-cells are sparsely active, with sparseness controlled by the hyperparameter $W$. S-cells, on the other hand, will typically be densely active, since they are linear.

Step 4 normalizes the $k$-dimensional feature vector, and also multiplies by the Euclidean norm of the input patch, to prevent the normalized output from being large even if the input is vanishingly small. The kernels in the layer are size $p \times p$, so that the input patch at any location contains $mp^2$ values where $m$ is the number of input feature maps.

**Final average pooling layer** To reduce the output dimensionality, we average pool each output feature map of the last energy layer to a $2 \times 2$ grid, exactly as in [8]. We have generally avoided pooling because we want to learn invariances rather than design them in. However, a final average pooling will turn out to be advantageous later on for speeding up meta-learning. In Table 2 we show that this pooling layer has only a modest impact on the final clustering accuracy.

# 4 Stacked unsupervised learning

The outputs of the ZCA layer are used as inputs to a convolutional energy layer. We train the kernels of this layer, and then we freeze the kernels. The outputs of the first convolutional energy layer are used as inputs to a second convolutional energy layer, and the kernels in this layer are trained. We repeat this procedure to stack a total of three convolutional energy layers, and then conclude with a final pooling layer.

| LAYER | # SUBSPACES ($k$) | SUBSPACE RANK ($r$) | # WINNERS ($w$) | KERNEL SIZE | PADDING |
|---|---|---|---|---|---|
| L1 | 37 | 2 | 9 | 8 | 2 |
| L2 | 9 | 3 | 8 | 5 | 1 |
| L3 | 58 | 16 | 2 | 21 | 2 |

Table 1: Detailed architecture of our three energy layer net. The first energy layer is preceded by convolutional ZCA whitening of the input image, where the kernel size is 9 and the number of whitened eigenvalues is 9. These parameters are found with automated hyperparameter tuning.

| REPRESENTATION | PIXELS | ZCA | LAYER 1 | LAYER 2 | LAYER 3 | 2X2 POOL |
|---|---|---|---|---|---|---|
| CLUSTERING ERROR (%) | 46.2 | 50.0 | 22.7 | 22.2 | 2.3 | 2.1 |

Table 2: Clustering error after every layer of our network with three energy layers.

**K-Subspaces clustering** Consider an energy layer with $k$ C-cells and $kr$ S-cells at each location. The S-cells at one location are linearly related to the input patch at that location by the set of $k$ matrices $\mathbf{V}_j$ for $j = 1$ to $k$, each of size $r \times mp^2$. Here $mp^2$ is the size of the flattened input patch, where $m$ is the number of input feature maps and $p$ is the kernel size.

We can think of these matrices as defining a set of $k$ linear subspaces of rank $r$ embedded in $R^{mp^2}$. We learn these matrices with a convolutional extension of the K-Subspace learning algorithm [45]. Let $\mathbf{x}_n$ be the previous layer's $m$-dimensional feature maps for pattern $n$. Define $\mathbf{x}_{n,i}$ as the $mp^2$-dimensional feature vector created by flattening a $p \times p$ patch centered around location $i$. K-Subspaces aims to learn $k$ $r$-dimensional subspaces $\mathbf{V}_k \in \mathbb{R}^{r \times mp^2}$ such that every patch is well modeled by one of these subspaces. This is formalized with the following optimization:

$$\min_{\mathbf{V}} \min_{\mathbf{C}} \sum_{n,i} \sum_k c_{nik} \left\| \mathbf{x}_{ni} - \mathbf{V}_k^\top \mathbf{V}_k \mathbf{x}_{ni} \right\|^2 \tag{1}$$

such that $c_{nik} \in \{0, 1\}$ and $\sum_k c_{nik} = 1$. We provide a novel minibatch algorithm for this optimization in Appendix A. During the learning, each matrix $\mathbf{V}_j$ is constrained so that its rows are orthonormal. Therefore at each location the C-cells contain the Euclidean norms of the projection of the input patch onto each of the subspaces, before the adaptive thresholding and rescaling. This is why the K-Subspaces algorithm is naturally well-suited for learning the convolutional energy layer.

During K-Subspaces learning, an input patch is assigned to a single subspace, which means there is winner-take-all competition between C-cells at a given location. During inference, on the other hand, there can be many C-cells active at a given location (depending on the "number of winners" hyperparameter $w$).

This idea of making representations less sparse for inference than for learning has been exploited by a number of authors [7, 24]. This may be advantageous because overly sparse representations can suffer from sensitivity to distortions [32].

**Experiments with MNIST digits** We train a network with the architecture of Fig. 1. The details of the ZCA layer and the three convolutional energy layers are specified by Table 1. Each energy layer is trained using a single pass through the 60,000 MNIST [27] training examples (without using the labels), with a minibatch size of 512. Training on a single Nvidia Titan 1080-Ti GPU takes 110 seconds.

**Evaluation of learned representations** We adopt the intuitive notion that the output representation vectors of a "good" network should easily cluster into the underlying object classes as defined by image labels. This is quantified by applying the trained network to 10,000 MNIST test examples. The resulting output representation vectors are clustered with the *scikit-learn* implementation of K-Means. This uses full batch EM-style updates and additionally returns the lowest mean squared error clustering found by running the algorithm using 10 different initializations.

We set the number of clusters to be 10, to match the number of MNIST digit classes. We compute the disagreement between the cluster assignments and image labels, and minimize over permutations of the clusters. The minimal disagreement is the final evaluation, which we will call the clustering error [49].

| REPRESENTATION | CLUSTERING ERROR (%) |
|---|---|
| PIXELS | 46.2 |
| NMF[‡] [28] | 44.0 |
| STACKED DENOISING AUTOENCODERS[†] [46] | 18.8 |
| UMAP [35] | 17.9 |
| GENERATIVE ADVERSARIAL NETWORKS[†] [12] | 17.2 |
| VARIATIONAL AUTOENCODER[†] [22] | 16.8 |
| DEEP EMBEDDED CLUSTERING[†] [48] | 15.7 |
| VaDE [19] | 5.5 |
| CLUSTERGAN[‡] [38] | 5.0 |
| UMAP + GMM [35] | 3.6 |
| N2D [34] | 2.1 |
| **OURS (THREE LAYER NET)** | **2.1** |
| INVARIANT INFORMATION CLUSTERING (AVG SUB-HEAD) [†] [18] | 1.6 |
| STACKED CAPSULE AUTOENCODERS [23] | 1.3 |
| INVARIANT INFORMATION CLUSTERING (BEST SUB-HEAD)[†] [18] | 0.8 |

Table 3: Comparison of clustering accuracy for other unsupervised learning algorithms. For methods which do not generate clusters, $k$-means is used to cluster representations into 10 clusters, with the exception of UMAP + GMM in which case we use a Gaussian Mixture Model to cluster. The errors for methods with a dagger [†] are all taken from [18], methods with double dagger [‡] are taken from [38]. UMAP uses "out-of-the-box" parameter settings.

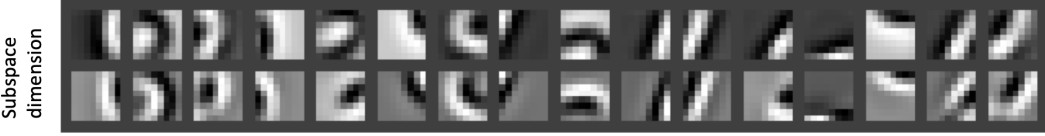

16 out of 37 learned 2D subspaces

Figure 3: Layer 1 subspaces learned with our algorithm. Each $8 \times 8$ image corresponds to a kernel.

Table 2 quantifies the accuracy of each layer. The accuracy of the ZCA representation is actually worse than that of the raw pixels. However, the accuracy of the representations improves with each additional convolutional energy layer, until the final error is just 2.1%.

Comparisons with other algorithms are shown in Table 3. It is helpful to distinguish between algorithms that require training data augmentation, and those that do not. Many well-known unsupervised algorithms that do not make use of training data augmentation, such as GANs, variational autoencoders, and stacked denoising autoencoders, yield clustering errors of 15 to 19%. Methods such as VaDE and ClusterGAN encourage clustered latent representations and these give rise to much lower clustering error. Clustering 2D UMAP representations with K-Means gives suprisingly high error, and this is likely the returned clusters are not spherical. Using a Gaussian Mixture Model instead gives much lower error. See the appendix for more discussion.

Self-supervised algorithms require training data augmentation, using prior knowledge to create same-class image pairs. For MNIST clustering one of the highest performance is Invariant Information Clustering, with a clustering error of 1.6 - 0.8% [18]. Our approach delivers roughly 2 % error and is noticeably better than the other algorithms that do not require training data augmentation.

Stacked capsule autoencoders [23] also achieve high accuracy. However, this algorithm incorporates a model of geometric distortions. Furthermore, despite the modifier "stacked," the algorithm does not conform to the original idea of repeatedly stacking the same learning module. The architecture uses several distinct types of layers and still backpropagates gradients.

**Learned kernels** Figure 3 shows that the kernels in the first energy layer look like bars or edges, more often curved than straight. Since the subspaces are of rank 2 (Table 1), the kernels come in pairs. The kernels in a pair look quite similar to each other, and typically appear to be related by small distortions. Therefore the two S-cells in a pair should prefer slightly different versions of the same feature. By computing the square root of the sum-of-squares of the S-cells, the C-cell should detect the same feature with more invariance to distortions.

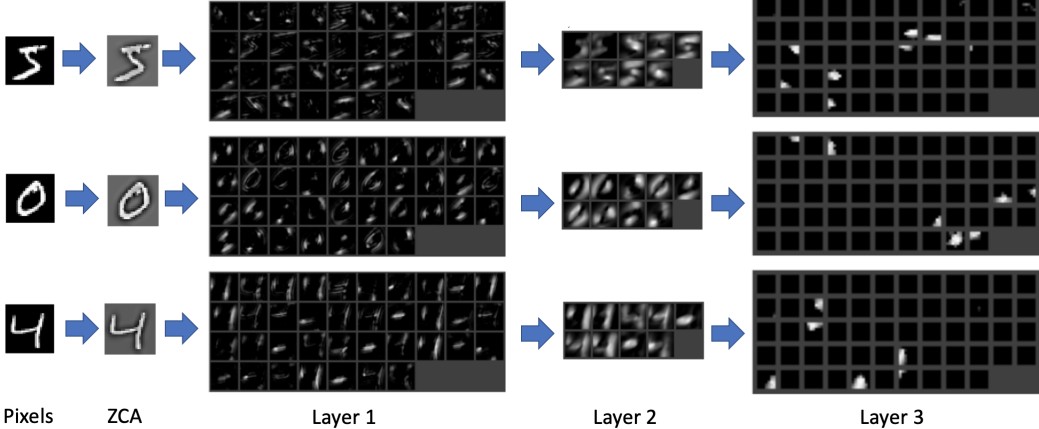

Figure 4: Output "C" feature maps for 3 input images.

This behavior is reminiscent of energy models of simple and complex cells in primary visual cortex [1, 42]. A complex cell computes the sum-of-squares of simple cells, which are quadrature pairs of Gabor filters. The sum-of-squares is invariant to phase shifts, much as the sum-of-squares of sine and cosine is constant. Our model also contains a sum-of-squares, but the filters are learned rather than hand-designed. Learning of complex cells was previously demonstrated by [16] for an energy model and by [20] for a similar model. [15] showed how to substitute rectification for quadratic nonlinearity. Our contribution is to stack multiple energy layers, and investigate the accuracy of the resulting network at a clustering task.

**Feature maps** Figure 4 shows the C-maps for the first 3 MNIST digits. The first energy layer exhibits an intermediate degree of sparsity, (approximately 20 % of the maps are active at central locations). The second energy layer exhibits dense representations (nearly all maps are active at central locations). The final energy layer shows quite sparse representations (approximately 5% of the maps are active). The feature maps appear to be broad spots.

## 5   Meta-learning of network architecture for unsupervised learning

The detailed architecture of the network is specified in Table 1, and one might ask where it came from. For example, the kernel size is 21 in the third energy layers, but sizes are only single digit integers in other layers. The subspaces are rank 2 in the first energy layer, but rank 3 and 16 in the subsequent energy layers. The second energy layer is highly dense (all but one neuron is active at each location), while the third layer is highly sparse (only two neurons are active at each location). There is a total of 17 numbers in Table 1, and we can regard them as hyperparameters of the unsupervised learning algorithm.

In the initial stages of our research, we set hyperparameters by hand, guided by intuitive criteria such as increasing the subspace rank with layer (meaning more invariances learned). Later on, we resorted to automated hyperparameter tuning. For this purpose, we employed the Optuna software package, which implements a Bayesian method [2]. We found that we were able to find hyperparameter configurations with considerably better performance than our hand-designed configurations.

In particular, the configuration of Table 1 was obtained by automated search through 2000 hyperparameter configurations, which took approximately 20 hours using 8 Nvidia GTX 1080 Ti GPUS. Each hyperparameter configuration was used to generate an unsupervised clustering of the training set, and its accuracy with respect to all 60,000 training labels was the objective function of the search.

For the ZCA layer we tune the kernel size $k_{zca} \in [1, 11]$ and number of whitened eigenvalues $n_{zca} \in [0, k_{zca}^2]$. For each subspace layer we tune number of subspaces $k \in [2, 64]$, subspace dimension $r \in [1, 16]$, number of winners $w \in [1, k]$, kernel size $k_s \in [1, \text{input\_size}]$, and padding $p_s \in [0, \text{floor}(k_s/2)]$. The stride is fixed at 1.

In some respects, the optimized configuration of Table 1 ended up conforming to our qualitative expectations. Sparsity and kernel size increased with layer, consistent with the idea of a feature

| NUMBER OF LABELS IN TRAINING SET | 10 | 30 | 50 | 100 | 500 | 5000 | 60000 |
|---|---|---|---|---|---|---|---|
| % MISCLASSIFIED (TEST) | 30.0 | 9.2 | 10.7 | 5.3 | 4.6 | 2.9 | 2.1 |
| % MISCLASSIFIED (TRAIN) | 0.0 | 0.0 | 0.0 | 3.0 | 2.8 | 2.9 | 2.5 |

Table 4: Label efficiency of our stacked learning algorithm. In each case, the algorithm has access to all 60K unlabeled images from the training set. What varies is the number of labels we use to evaluate each setting of learning parameters.

| EXPERIMENT | % ERROR (TEST) | % ERROR (TRAIN) |
|---|---|---|
| 4 ENERGY LAYERS | 3.1 | 3.6 |
| 3 ENERGY LAYERS | **2.1** | **2.5** |
| 2 ENERGY LAYERS | 2.9 | 3.3 |
| 1 ENERGY LAYERS | 20.2 | 20.7 |
| NO ZCA - 3 ENERGY LAYERS | 4.0 | 4.8 |
| NO RESCALING - 3 ENERGY LAYERS | 4.1 | 4.6 |
| NO ZCA & NO RESCALING - 3 ENERGY LAYERS | 9.7 | 10.1 |
| 1D SUBSPACES - 3 ENERGY LAYERS | 16.0 | 17.3 |
| RANDOM SUBSPACES - 3 ENERGY LAYERS | 29.4 | 31.1 |

Table 5: Systematic studies with our multilayer energy model. The learning hyperparameters are tuned using all 60K labels from the training set.

hierarchy with progressively greater selectivity and invariance. However, the number of subspaces behaved nonmonotonically with layer, which was unexpected.

We can think of the hyperparameter tuning as an outer loop surrounding the unsupervised learning algorithm. We will refer to this outer loop as meta-learning. Given an architecture, the unsupervised learning algorithm requires no labels at all. However, the outer loop searches for the optimal network architecture by using training labels. Therefore, while the learning is unsupervised, the meta-learning is supervised. Alternatively, the outer loop can use only a fraction of the training labels, in which case the meta-learning is semi-supervised.

It is interesting to vary the number of training labels used for hyperparameter search. The results are shown in Table 4. The best accuracy is obtained when all 60,000 training set labels are used. Accuracy degrades slightly for 5000 labels, and more severely for fewer labels than that. The test error can be lower than the training error. This is not a mistake, and it appears to result from a non-random ordering of patterns in the MNIST train and test sets.

To be clear about the use of data, we note that neither test images nor labels are used during learning or meta-learning. Training images but not labels are used during learning. Training labels are used by meta-learning. With each iteration of meta-learning, the weights of the network are randomly initialized.

# 6 Experiments

The hyperparameter search explores the space of networks defined by Fig. 1. We can widen the space of exploration by performing ablation studies, with results given in Table 5. In all experiments, we completely retune the hyperparameters using the full 60K training labels to evaluate clustering accuracy. With the exception of the experiment where we vary the number of layers, we use the 3 energy layer network in this section.

**Vary number of energy layers** One can vary the depth of the network by adding or removing energy layers. 2 and 4 energy layers yield similar accuracy, and are roughly 1% absolute error (50% relative error) worse than for 3 energy layers. A 1 energy layer net is dramatically worse (>20% train/test error), suggesting the stacking is critical for performance.

The hyperparameters for each of these optimal architectures are provided in the Appendix. The optimal 2 energy layer net resembles the 3 energy layer net with its 2nd energy layer removed, while simultaneously making the 1st energy layer less sparse.

**Remove ZCA whitening** Removing the ZCA layer increases both train and test error of the three energy layer net by roughly $2\times$. This might seem surprising, as Table 2 shows that whitening by itself decreases accuracy if we directly cluster whitened pixels instead of raw pixels. Apparently, it is helpful to "take a step backward, before taking 3 steps forward" in the case of our networks with three energy layers. We have no theoretical explanation for this interesting empirical fact.

Preprocessing images by whitening has a long history. The retina has been said to perform a whitening operation, which is held to be an "efficient coding" that reduces redundancy in an information theoretic sense [4, 3]. Our experiment suggests that whitening is useful because it improves the accuracy of subsequent representations produced by stacked unsupervised learning. This seems at least superficially different from efficient coding theory, because the invariant feature detectors in our networks appear to discard some information (Figure 4).

**Remove rescaling** We now remove the rescaling operation from our energy layers. We observe a roughly $2\times$ increase in both test and train error. It may be surprising that such a seemingly innocuous change can double the resulting train/test errors. This is likely because we only have 17 hyperparameters to optimize; with a limited set of parameters to tune, small changes in architecture can lead to dramatic performance changes as we only have limited flexibility to tune hyperparameters.

**Remove rescaling and ZCA whitening** Removing both ZCA and the per-layer rescaling operations causes a $4\times$ increase in train and test error. Apparently the damage to the resulting performance is multiplicative: removing ZCA alone doubles the train/test error, removing rescaling alone doubles train/test error, and removing both quadruples the train/test error.

**1D subspaces** We rerun the automated hyperparameter tuning experiments from the previous section, this time restricting our subspaces to be 1D. The subspace norm can be thought of as a conventional dot product followed by absolute value nonlinearity $\|\mathbf{V}\mathbf{x}\| = |\mathbf{v} \cdot \mathbf{x}|$ where $\mathbf{v}$ is the one row of the subspace matrix $\mathbf{V}$. When the subspaces are 1D, our algorithm in fact reduces to that of [9].

**Random subspaces** Finally we ask the question: does learning actually help or is it simply the architecture that matters? To do so, we run tuning experiments with random orthogonal subspaces. We observe that performance is almost completely erased by using random subspaces, telling us that the unsupervised learning component is indeed critical for clustering performance.

# 7 Discussion

The elements of our SUL algorithm were already known in the 2000s: ZCA whitening, energy layers, K-Subspaces, and adaptive thresholding and rescaling of activities during inference. To achieve state-of-the-art unsupervised clustering accuracy on MNIST, we employed one more trick, automated tuning of hyperparameters. Such meta-learning is more feasible now than it was in the 2000s, because computational power has increased since then.

Given that meta-learning optimizes a supervised criterion, is our SUL algorithm really unsupervised? It is true that the complete system is supervised. However, even if the outer loop (meta-learning) is supervised, it is accurate to say that the inner loop (learning) is unsupervised. For any hyperparameter configuration, the network starts from randomly initialized weights, and proceeds to learn in a purely label-free unsupervised manner.

Although MNIST is an easy dataset by today's standards, we think that the success of our SUL algorithm at unsupervised clustering is still surprising. We are only tuning 17 hyperparameters, and it is not clear a priori that our approach would be flexible enough to succeed even for MNIST.

In future work, it will be important to investigate more complex datasets or tasks. Following the more common scenario for meta-learning, future work should train an unsupervised algorithm on a distribution of tasks, and test transfer to some held-out distribution. We have done some preliminary experiments with the CIFAR-10 dataset. The meta-learning is more time-consuming because larger network architectures must be explored. The research is still in progress, but we speculate that the winner-take-all behavior of K-Subspaces learning may turn out to be a limitation. If so, it will be important to relax the winner-take-all condition in Equation (1).

Evolution created brains through eons of trial-and-error. For us to discover how the brain learns, it will be important to exploit our computational resources, and use meta-learning to empirically search the space of biologically plausible learning algorithms.

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

# A  Online minimization of Eq. 1

---
**Algorithm 1:** K-Subspace clustering via minibatch power iterations

---
Initialize subspaces $\{\mathbf{V}_k \in \mathbb{R}^{r \times mp^2}\}$
**for** $i = 1, 2, 3$ **do**

    Sample patches from a minibatch of images $\{\mathbf{x}_u \in \mathbb{R}^{mp^2}\}$
    Cluster patches

$$\mathbf{X}_k := \{\mathbf{x}_u : k = \underset{q}{\operatorname{argmin}} \left\| \mathbf{x}_u - \mathbf{V}_q^\top \mathbf{V}_q \mathbf{x}_u \right\| \} \quad \text{for} \quad k = 1, 2, 3, \ldots$$

    Apply one orthogonal power iteration to every subspace

$$\mathbf{U}\mathbf{S}\mathbf{V}^\top := \mathbf{X}_k^\top \mathbf{X}_k \mathbf{V}_k^\top \qquad \text{SVD decomposition}$$

$$\mathbf{V}_k := \mathbf{U}^\top \quad \text{for} \quad k = 1, 2, 3, \ldots$$

**end**

---

A standard algorithm for minimizing Eq. 1 is a full batch EM algorithm that alternates between cluster assignment (E-step) and using PCA to set $\mathbf{V}_k$ to the top $r$ principle components of the patches assigned to each cluster $k$ (M-step) [45]. The full batch requirement makes this algorithm rather slow.

The algorithm we present and use in this paper is described in Algorithm 1. We make two core changes to the standard EM algorithm. One, we use minibatch updates instead of full batch updates. Two, we perform a single step of power-iteration after each cluster assignment step, instead of performing a full PCA.

Proper initialization can impact on the quality of learned subspaces. We initialize subspaces by setting the first dimension to a randomly chosen patch, and the other dimensions to white Gaussian

noise with $\mu = 0, \sigma = 0.01$. We then perform a few "warmup" iterations of the main loop in alg. 1, except we only cluster using the first subspace dimension. We use $warmup\_iter = 10$ in all our experiments. During these warmup iterations, the power updates are still performed on the full rank-$r$ subspaces. Intuitively, this warmup procedure generates clusters with 1D subspace clustering, and initializes subspaces be the top $r$ components within these clusters.

Convergence of our algorithm is discussed in the Appendix. We prove that the clustering+power iteration step ensures the loss computed over the minibatch is non-decreasing. We show empirically that the loss computed over the whole dataset decreases with iteration for a reasonable setting of parameters. Because we only use a single pass through the data to train each layer, our algorithm is much faster than applying the full batch EM-style algorithm described in [45] to the whole dataset.

## B  Convergence analysis of Algorithm 1

### B.1  Theory: full-batch convergence

We will not be able to provide a full proof that Algorithm 1 converges in the minibatch setting. However we can at least show that in the full batch setting, every iteration of Algorithm 1 decreases the energy in Equation (1). Of course the cluster assignment portion of the algorithm decreases the energy. What we will show here is that the power iteration step also decreases the energy at every iteration.

We recall the notation from Algorithm 1. We define the matrix $\mathbf{X}_k$ whose rows are the input patches assigned to cluster $k$:

$$\mathbf{X}_k := \{\mathbf{x}_u : k = \operatorname*{argmin}_q \|\mathbf{x}_u - \mathbf{V}_q^\top \mathbf{V}_q \mathbf{x}_u\|\} \tag{2}$$

The energy in Equation (1) can be written as a sum of energies for each cluster $e = \sum_k e_k$ where:

$$e_k = \|\mathbf{X}_k - \mathbf{X}_k \mathbf{V}_k^\top \mathbf{V}_k\|_F^2 \tag{3}$$

We will show that a power update for cluster $k$ now gives a non-decreasing energy $e_k$. To avoid notational clutter, we will drop the subspace index $k$ and assume we are working with a single fixed cluster for now. It will actually be easier to show that a more general class of updates than the SVD update in Algorithm 1 cause the energy to remain or decrease. Suppose we have any subspace update defined by:

$$\mathbf{V}' := \mathbf{Q}(\mathbf{V}\mathbf{C}\mathbf{V})^\dagger \mathbf{C}\mathbf{V} \tag{4}$$

where $\mathbf{Q}$ is any orthogonal $r \times r$ matrix (that can also be a function of $\mathbf{V}, \mathbf{C}$. The power update in Algorithm 1 is an example of such an update. We will show that the energy for every subspace is non-decreasing with this update: $e' \leq e$. We do so by relating the update in Equation (4) to one sequence of steps in an alternating least squares problem. Define:

$$h(\mathbf{A}, \mathbf{B}) := \|\mathbf{X} - \mathbf{A}\mathbf{B}^\top\|^2 \tag{5}$$

Define $\mathbf{A} = \mathbf{X}\mathbf{V}^\top$ and $\mathbf{B} = \mathbf{V}^\top$. Then $e = h(\mathbf{A}, \mathbf{B})$. Define the sequence:

$$\begin{aligned}
\mathbf{B}' &= \operatorname*{argmin}_{\mathbf{U}} h(\mathbf{A}, \mathbf{U}) = \mathbf{X}\mathbf{A}(\mathbf{A}^\top \mathbf{A})^\dagger \\
\mathbf{A}' &= \operatorname*{argmin}_{\mathbf{U}} h(\mathbf{U}, \mathbf{B}') = \mathbf{X}\mathbf{B}'((\mathbf{B}')^\top \mathbf{B}')^\dagger
\end{aligned} \tag{6}$$

We can multiply $\mathbf{A}'(\mathbf{B}')^\top$ and substitute the above equations to get:

$$\mathbf{A}'(\mathbf{B}')^\top = \mathbf{X}(\mathbf{V}')^\top \mathbf{V}' \tag{7}$$

We therefore have that $e' = h(\mathbf{A}', \mathbf{B}') \leq h(\mathbf{A}, \mathbf{B}) = e$, so the subspace energy is non-increasing under the updates in Algorithm 1.

### B.2  Experiment: learning curves

We show learning curves using Algorithm 1 applied to MNIST digits in Figure 5. We run this algorithm for a single pass through the 60k training set patterns. Inputs are first whitened with

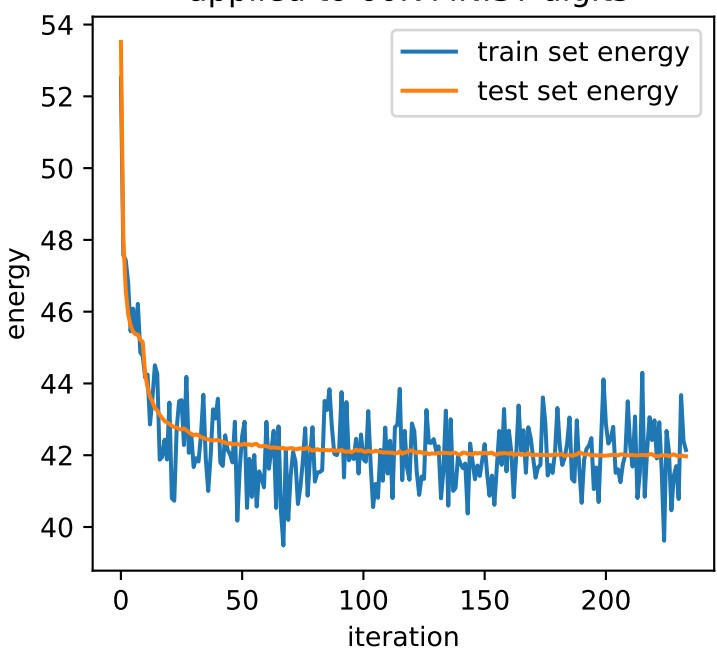

Figure 5: Learning algorithms for Algorithm 1 applied to MNIST digits.

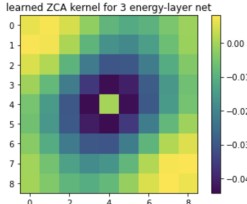

Figure 6: Learned ZCA kernel for 3 energy layer net. We have zero-ed out the central pixel, as its value is 1.17 and thus would make it harder to see the surround structure. This filter resembles an oblique center surround filter.

ConvZCA using a kernel size of 11 and num component of 16. We learn 64 subspaces, each of dimension 5, and kernel size of 9. We use a minibatch size of 256.

The training set energy is computed over a minibatch of 256 inputs at each iteration. The test set energy is computed over all 10K test set patterns, which is why it is less noisy. Empirically we see that for this setting of parameters at least, our minibatch K-Subspace algorithm does lead to a decreasing energy computed over unseen patterns.

We apply 10 warmup iterations (only using the first subspace dimension to cluster for the first 10 iterations), which is why we observe the sharp drop-off in energy after 10 iterations.

## C  Hyperparameters for 1,2,3,4 energy layer nets

In Table 6 we show the learned architectures for the 1,2,3,4 energy layer networks.

Table 6: Network architectures for 1,2,3,4 energy layer nets from Table 5.

| HYPERPARAMETER | 1 LAYER | 2 LAYER | 3 LAYER | 4 LAYER |
|---|---|---|---|---|
| CONVZCA KERNEL SIZE | 5 | 9 | 9 | 11 |
| CONVZCA N COMPONENTS | 0 | 18 | 9 | 18 |
| LAYER1 SUBSPACE NUMBER ($k$) | 59 | 20 | 37 | 15 |
| LAYER1 SUBSPACE RANK ($r$) | 2 | 5 | 2 | 2 |
| LAYER1 ACTIVE FEATURES ($w$) | 1 | 16 | 9 | 10 |
| LAYER1 KERNEL SIZE | 10 | 10 | 8 | 7 |
| LAYER1 PADDING | 4 | 2 | 2 | 3 |
| LAYER2 SUBSPACE NUMBER ($k$) | - | 55 | 9 | 63 |
| LAYER2 SUBSPACE RANK ($r$) | - | 16 | 3 | 12 |
| LAYER2 ACTIVE FEATURES ($w$) | - | 1 | 8 | 1 |
| LAYER2 KERNEL SIZE | - | 19 | 5 | 17 |
| LAYER2 PADDING | - | 1 | 1 | 1 |
| LAYER3 SUBSPACE NUMBER ($k$) | - | - | 58 | 57 |
| LAYER3 SUBSPACE RANK ($r$) | - | - | 16 | 7 |
| LAYER3 ACTIVE FEATURES ($w$) | - | - | 2 | 6 |
| LAYER3 KERNEL SIZE | - | - | 21 | 8 |
| LAYER3 PADDING | - | - | 2 | 2 |
| LAYER4 SUBSPACE NUMBER ($k$) | - | - | - | 22 |
| LAYER4 SUBSPACE RANK ($r$) | - | - | - | 1 |
| LAYER4 ACTIVE FEATURES ($w$) | - | - | - | 1 |
| LAYER4 KERNEL SIZE | - | - | - | 3 |
| LAYER4 PADDING | - | - | - | 1 |

## D    Learned ZCA filter

In Figure 6 we show the learned ConvZCA kernel for the 3 energy layer network. It resembles an oblique center surround filter. It is interesting to calculate the relative weight of the negative surround vs positive center term. Specifically we calculate:

$$f = \frac{I[4,4] - \sum_{u,v} \max\{0, -I_{u,v}\}}{I[4,4]} = 0.10 \tag{8}$$

where $I$ is the 9x9 kernel and $I[4,4]$ is the central pixel of that kernel. In other words, there is a small DC component (the central pixel is not perfectly cancelled out by the negative surround).

## E    Clustering UMAP embeddings

In the main text we reported a surprisingly low performance for K-Means applied to UMAP embedding, and that using a Gaussian mixture model instead can significantly improve the accuracy. Here we show the UMAP embeddings, and corresponding clusterings generated via K-Means and Gaussian Mixture Models.

Note that our result is not inconsistent with the result given in Table 1 of [34] who reported 82.5% clustering accuracy (compared to our result of 96.4% in Table 5) when using a GMM to cluster the embedding vectors return by UMAP. This is because we are using UMAP to embed to 2D while they used UMAP to embed to 10 dimennsions. For this task it seems that the extremely low dimensional embeddings are actually more suitable for downstream clustering.

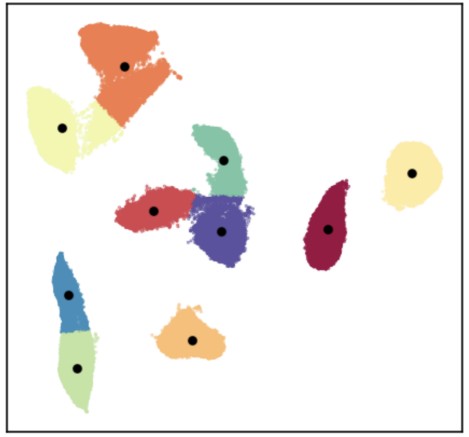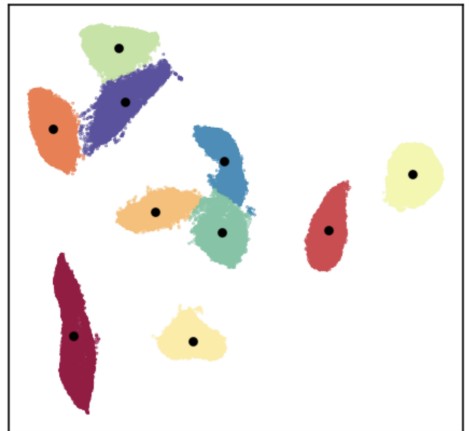

Figure 7: K-Means vs. Gaussian Mixture-based clustering applied to UMAP with "out-of-the-box" parameters applied to MNIST handwritten digits. Colors are assigned to each point based of the clustering (not the ground truth labels). K-Means erroneously merges and splits some of the clusters, while Gaussian Mixture models give a much more intuitive clustering (and ultimately a much lower clustering error as shown in Table 5)

