# OpenReview forum: "Stacked unsupervised learning with a network architecture found by supervised meta-learning"
_NeurIPS.cc/2022/Conference — NeurIPS 2022 Submitted_

### Official Review · Reviewer_CyHx · 2022-07-06

**Rating:** 6
**Confidence:** 4
**Soundness:** 4 excellent
**Presentation:** 3 good
**Contribution:** 3 good

**Summary:**

In this manuscript the authors create an architecture and learning scheme to learn the clustering of MNIST without backpropagation. They combine energy layers, which compute responses as vector lengths in subspaces, with adaptive thresholding that introduces a form of competition between the output features. These layers are then learned consecutively to optimize K-Subspaces clustering, i.e. to minimize the distance of samples from their closest subspace. For this procedure the authors give a new mini-batched algorithm. After applying a supervised architecture search to this approach the authors find an architecture that reaches 2% error rate on MNIST.

**Questions:**

1) Could you please add some discussion on what the conclusions for future work on unsupervised learning are? Solving MNIST is not really an aim in itself.
2) What is the authors opinion on biological plausibility here?
3) And how is the interpretation of the supervised meta-learning on top of unsupervised learning results different from unsupervised learning only?


**Limitations:**

---

**Strengths And Weaknesses:**

Constructing this learning system is definitely an engineering feat, but it is unclear to me what the novel insight we are supposed to gain from this manuscript. From a cautionary tale like “even fitting only the architecture we can achieve much better performance with somewhat unexpected parameters” to “this is a great architecture that may generalize to much harder unsupervised learning tasks” many interpretations would be possible and the authors do not comment on this and I don’t think their data are sufficiently clear on this on their own. Nonetheless, the authors are thorough. For example, they investigate the convergence of their batch algorithm, find a working clustering for the UMAP embedding and check the necessity of the parts of their model.

Concrete points:
1) It remains a bit unclear what the authors relationship to biological plausibility is here. They emphasize that stacked unsupervised learning is more plausible than back propagation, but other aspects like the subspace clustering for example are not questioned regarding back propagation at all.
2) Somewhat surprisingly, the authors find that 4 layers work less well than 3 layers on MNIST. While choosing 3 layers then is consistent with the architecture search theme, this seems odd. Is there an explanation why deeper features are no longer better?
3) While I tend to agree that supervised learning of the architecture for a task does not necessarily break biological plausibility, I don’t think all advantages of unsupervised learning still apply. Practically for example, the method still requires that supervised data is available for meta-learning. I think some discussion of this is necessary.

---

> ### Author Response · Authors · 2022-08-02
> **Reply to CyHx**
>
> *It remains a bit unclear what the authors relationship to biological plausibility is here. They emphasize that stacked unsupervised learning is more plausible than back propagation, but other aspects like the subspace clustering for example are not questioned regarding back propagation at all.*
>
> a. We regret that we did not explain the historical context properly. There has been a great deal of research showing that single layer networks with Hebb-like synapses can implement unsupervised learning algorithms like independent component analysis, independent subspace analysis, clustering, principal component analysis, nonnegative matrix factorization, etc. Some work is required to extend such research to subspace clustering, but the extension would not be surprising. (Note that Hebbian implementation of the closely related independent subspace analysis is already discussed by Hyvarinen and Hoyer 2000.) Because such networks are biologically plausible, stacking them is also biologically plausible (the SUL approach), and has been attempted since around 1980. Unfortunately, the performance of SUL has been so poor in practical applications that SUL seems untenable as a brain theory. This is why our work focuses on stacking as the most important issue rather than the precise biological implementation of subspace clustering.
>
> b. The above remarks focus on the biological plausibility of our model of learning in a single lifetime. We also think that the the key biological insight is that biological learning may be largely unsupervised as an animal does not seem to have access to a large amount of labels during its lifetime. However, via evolution, learning rules may be encoded which implicitly contain information.
>
> c. You are correct however, that we do not know how subspace clustering could be implemented in a biological network
>
> *Somewhat surprisingly, the authors find that 4 layers work less well than 3 layers on MNIST. While choosing 3 layers then is consistent with the architecture search theme, this seems odd. Is there an explanation why deeper features are no longer better?*
>
> We were also surprised by this finding. Our initial expectation was that 4 layers can’t be worse than 3 layers, because the fourth layer could just be the identity function. But actually this reasoning was faulty. Because the weights are set by the learning algorithm, the fourth layer cannot be the identity function. Another possibility is that the hyperparameter search has somehow failed to find the optimal solution. We chose the Optuna package for hyperparameter search because it was convenient; it’s possible that better algorithms and/or software exist.
>
> *While I tend to agree that supervised learning of the architecture for a task does not necessarily break biological plausibility, I don’t think all advantages of unsupervised learning still apply. Practically for example, the method still requires that supervised data is available for meta-learning. I think some discussion of this is necessary.*
>
> We would argue that the supervised learning of architecture is one of the reasons this is biologically plausible. But yes we agree, because we need labels to learn, this algorithm in its current form really does not have any of the advantages of a standard unsupervised learning algorithm. Future work should investigate transfer of the learning rules (the ideal would be learning the learning rules on imagenet, then applying the learned rules to a novel dataset)
>
> *Could you please add some discussion on what the conclusions for future work on unsupervised learning are? Solving MNIST is not really an aim in itself.*
>
> The conclusion is that by viewing evolution as an outer supervised learning loop with unsupervised learning on the inner loop of meta learning, we may be able to significantly advance the performance of previous bio-inspired learning methods.
>
> *What is the authors opinion on biological plausibility here?*
>
> There are two sources of biological plausibility. One is the fact that unsupervised learning happens with communication local to each layer (so no backprop). Two, we view evolution as a supervised method to tune unsupervised learning rules. Biologically plausible implementations of subspace clustering can be constructed, similar to implementations of independent subspace analysis, but we did not discuss this in the paper because we regard stacking as the central issue.
>
> *And how is the interpretation of the supervised meta-learning on top of unsupervised learning results different from unsupervised learning only?*
>
> The key point is that by using supervised learning to tune the unsupervised learning rules, we were able to stack well known unsupervised-learning modules (whitening, subspace clustering) and generate a surprisingly accurate clustering. Without the supervised learning, it is essentially very hard to tune the hyperparameters by hand.

---

### Official Review · Reviewer_x8sP · 2022-07-09

**Rating:** 5
**Confidence:** 4
**Soundness:** 2 fair
**Presentation:** 3 good
**Contribution:** 2 fair

**Summary:**

This paper addresses the issue of how to design unsupervised learning(UL) architecture and learning without backpropagation that matches the performance of UL with backpropagation. It proposes an UL architecture consisting of a whitening layer, multiple convolutional energy layers and a final average pooling layer. The  convolutional kernels are learned with K-Subspace algorithm, and the hyperparameters are chosen by supervised meta learning to optimize the clustering accuracy. Experiments on MNIST dataset demonstrates that the proposed method achieves comparable accuracy as UL with backpropagation.

**Questions:**

The experiments are limited. It evaluates only on MNIST which is an easy dataset. Experiments on more complex datasets/tasks will help to support the claims.


**Limitations:**

It discussed the limitations of the work, but did not discuss potential negative societal impact of the work.

**Strengths And Weaknesses:**

Strengths:
+ The inner loop of the proposed algorithm has analogous to brain, and can train the model without backpropagation.

+ The authors conducted extensive ablation study indicating the contribution of each proposed components.

+ The related work section has a nice flow and puts the proposed method into context.

Weaknesses:
- Winner-take-all of K-subspace learning does not guarantee the learned convolutional kernels are optimal.

- Searching for optimal hyperparamters requires supervised learning, which makes the proposed method not purely unsupervised.

- The experiments are limited. It evaluates only on MNIST which is an easy dataset.

---

> ### Author Response · Authors · 2022-08-02
> **Reply to x8sP**
>
> *Winner-take-all of K-subspace learning does not guarantee the learned convolutional kernels are optimal.*
>
> This is correct. However, this statement is true of nearly every deep network as well: we have no guarantees that they actually find a global minima of the object functions they are trained on.
>
> *Searching for optimal hyperparameters requires supervised learning, which makes the proposed method not purely unsupervised.*
>
> a. This is also correct. However, this is precisely the behavior we wish to model. Biological learning seems largely unsupervised as an animal does not seem to have access to a large amount of labels during its lifetime. However, via evolution, learning rules may be encoded which implicitly contain information from generations of past lives that sum up to using large amounts of labels.
>
> b. It should also be noted that the supervisory signal is extremely weak. It is a single scalar computed from the training labels, rather than a gradient for all the weights.
>
> c. Our setting (unsupervised inner loop, supervised outer loop) is novel. It is not straightforward to compare our setting with other settings such as unsupervised, few-shot, or semi-supervised. We chose to compare with unsupervised clustering, because the weights of our network contain no information derived from labels; that information resides only in the hyperparameters. In few-shot or semi-supervised settings, the weights of the network contain information from labels.
>
> *The experiments are limited. It evaluates only on MNIST which is an easy dataset.*
>
> This is correct. Even achieving success on MNIST was quite involved. We hope this success can inspire future papers to work on more involved datasets.

---

> > ### Comment · Reviewer_x8sP · 2022-08-06
> > **Keep Score**
> >
> > Thanks for the detailed answer. Most of my concerns are addressed. After reading the other reviews and the answers, I confirm my rating.

---

### Official Review · Reviewer_WR6r · 2022-07-11

**Rating:** 5
**Confidence:** 3
**Soundness:** 3 good
**Presentation:** 3 good
**Contribution:** 2 fair

**Summary:**

This paper designed a stacked unsupervised learning (SUL) architecture to learn visual classification task without backpropagation. The proposed SUL has two loops. The inner loop is unsupervised learning, which trained the kernel of each layer iteratively, and thus no backpropagation was required. The outer loop is supervised learning to optimize the hyperparameters of the SUL. The proposed SUL was tested on Mnist dataset and achieved a good result compared to other unsupervised learning methods.

**Questions:**

1. Did other unsupervised learning methods also fine-tune their hyper parameters? Did other methods also require the labels of the images? If not, it seems unfair to compare the performance of the current method to traditional unsupervised learning method. It may be better to compare the current SUL to some few-shot learning or semi-supervised methods.
2. Why does the automated tuning of hyperparameters affect the accuracy significantly?
3. Will SUL also work on other more complex datasets?
4. What's the theoretic contribution of the designed SUL?
5. Besides biologically plausible, are there any other advantages for SUL compared to backpropagation? Will it achieve higher accuracy than backpropagation? Will it be more efficient to train? Will it explain some biological problem better? Or will it be more feasible on some special applications?
6. As mentioned in the paper, the meta-learning is more time-consuming on CIFAR-10 dataset. Considering that the training process still requires labels. What is the advantage of SUL compared to supervised learning or few-shot learning?

**Strengths And Weaknesses:**

Strengths:
1. This paper illustrated that SUL can still work on a visual classification task without backpropagation.
2. This paper achieved a state-of-art performance on an unsupervised learning task.

Weaks:
1. All items, including ZCA, energy layers, K-subspaces, and adaptive thresholding, were already proposed in the 2000s. This paper only introduced automated tuning of hyper-parameters, which seems just a marginal experimental contribution.
2. Because the proposed method still requires many labels (~5000) for meta-learning, it seems unfair to compare the current method with the previous unsupervised learning method.
3. This paper lacks a theoretical explanation about why the current SUL works but SULs back in the 2000s did not work well.

---

> ### Author Response · Authors · 2022-08-02
> **Response to WR6r**
>
> *Did other unsupervised learning methods also fine-tune their hyper parameters?*
>
> a. Our setting (unsupervised inner loop, supervised outer loop) is novel. It is not straightforward to compare our setting with other settings such as unsupervised, few-shot, or semi-supervised. We chose to compare with unsupervised clustering, because the weights of our network contain no information derived from labels; that information resides only in the hyperparameters. In few-shot or semi-supervised settings, the weights of the network contain information from labels.
>
> b. It should also be noted that the supervisory signal is extremely weak. It is a single scalar computed from the training labels, rather than a gradient for all the weights.
>
> c. Researchers on unsupervised clustering are likely performing some model selection based on the test error, though this is not explicitly described in the literature. (An exception is the cited paper on Information Invariant Clustering, where one of the accuracy numbers is for the best “sub-head” of a network with multiple subheads.) Certainly the field as a whole is engaging in model selection based on the test error.
>
> d. Other methods do not explicitly require the labels of images. However, self-supervised methods generally depend on training set augmentations that produce pairs of images that belong to the same class. Our method does not make use of such augmentations.
>
> e. Note that our goal is not to find a method that exceeds the state-of-the-art in practical applications. Rather, our goal is to show that SUL, which is thought to have awful performance based on the previous studies of Table 3, can actually be comparable to state-of-the-art methods if done properly (at least for MNIST).
>
> f. We are interested in SUL because of its potential as a theory of unsupervised learning in the brain. Also the setting we propose (SUL inside supervised meta-learning outer loop) can be interpreted as a model of brain evolution.
>
> *Why does the automated tuning of hyperparameters affect the accuracy significantly?*
>
> a. The tuning of hyperparameters might be dismissed as a “marginal experimental contribution,” because it sounds trivial in retrospect. Actually it took us a long time to think of doing this, because we (and previous researchers) always assumed that accuracy would depend only weakly on hyperparameters. There are not so many hyperparameters, and they control rather innocuous-looking aspects of the model such as network size and sparsity of activation. In the end, we were surprised to find that accuracy does depend rather strongly on hyperparameter tuning. This is not a theoretical insight, but it is a surprising empirical finding that has the potential to lead to interesting theoretical work.
>
> b. This sensitivity to hyperparameters is presumably the reason why the current SUL works much better than SUL from the 2000s. This indeed merits theoretical explanation. However, finding a theoretical explanation seems nontrivial, so we regard it as a subject for future research.
>
> c. We speculate that SUL may be more sensitive to hyperparameters than self-supervised learning algorithms that make use of training data augmentations. We are not sure, however, and more research is needed concerning this issue.
>
> *Will SUL also work on other more complex datasets?*
>
> This is a very interesting question! While the final result of our paper is easy to understand, it was actually quite challenging to arrive at such a simple result. This success was crucial for convincing ourselves that SUL is worth exploring further in the future, and we hope that our paper will also motivate other researchers to try more complex datasets.
>
> *What's the theoretic contribution of the designed SUL?*
>
> a. We think that obtaining decent performance with SUL, even on MNIST, is a surprising result, given that the conventional wisdom that SUL is basically useless due to the poor accuracies cited in Table 3. This surprising empirical finding has the potential to motivate interesting theoretical explorations.
>
> b. The setting (unsupervised inner loop, supervised outer loop) is of theoretical interest because it can be conceptualized as a combination of learning and evolution.

---

> > ### Author Response · Authors · 2022-08-02
> > **Continuation of reply to WR6r**
> >
> >
> > *Besides biologically plausible, are there any other advantages for SUL compared to backpropagation? Will it achieve higher accuracy than backpropagation? Will it be more efficient to train? Will it explain some biological problem better? Or will it be more feasible on some special applications?*
> >
> > Again we emphasize that our main goal is brain theory. We are not aiming to beat backpropagation, which is an extremely versatile and powerful engineering technique. That being said, one could imagine that SUL has some engineering advantages. It is likely premature to speculate, given that we have not yet demonstrated success with datasets more complex than MNIST. Nevertheless, we feel emboldened to speculate given the request of the reviewer. One could imagine that SUL is more robust to adversarial examples than nets trained by vanilla backprop. SUL might also handle domain adaptation or nonstationarity well.
> >
> > *As mentioned in the paper, the meta-learning is more time-consuming on CIFAR-10 dataset. Considering that the training process still requires labels. What is the advantage of SUL compared to supervised learning or few-shot learning?*
> >
> > We ultimately hope that this class of methods will generalize across datasets. So one would be able to to meta-learn on a dataset like ImageNet, and then apply the learned learning rules to a novel dataset. This remains to be demonstrated though.

---

> > > ### Comment · Reviewer_WR6r · 2022-08-08
> > > **Increase score to 5**
> > >
> > > Thank authors for the detailed responses, which addressed most of my questions. I would like to increase the rate to 5.

---

### Public Comment · ~Celestine_Preetham_Lawrence1 · 2022-11-11
**Insight on hyperparameter landscape**

Can we learn something from your hyperparameter landscape? Are the optimal ones more biologically abundant than the suboptimal ones?
Are the extrema thin or broad ? Your work has potentially many fascinating implications for evolutionary biology (which the NeurIPS reviewers seem to have unfortunately missed).

---

### Meta-Review · Area_Chair_iLD3 · 2022-08-27

**Recommendation:** Reject
**Confidence:** Certain

**Metareview:**

This paper proposes a specific architecture for performing stacked unsupervised learning, and demonstrates this algorithm on mnist.

All reviewer scores are borderline, and one reviewer lowered their score during discussion. No reviewer was willing to champion the paper during discussion. Based upon this, I recommend rejection.

I spent a little bit of time looking at the paper myself. One particular concern I have from my own reading is that the proposed algorithm was extensively meta-trained on the target task (MNIST classification), so it wasn't clear to me that this should count as an unsupervised algorithm. (More typically, meta-training would be performed on other tasks, and then the learned algorithm applied to the target task.) (Reviewer x8sP made a similar observation.)

**Award:**

No

---

### Decision · Program_Chairs · 2022-09-14

Reject